# Convergence of Conceptual Innovation Model to Reduce Challenges Faced by the Small and Medium Sized Enterprises' (SMEs) in Bangladesh

**Tarnima Warda Andalib \*** **and Hasliza Abdul Halim**

School of Management, Universiti Sains, Pulau Pinang 11800, Malaysia
* Correspondence: tarnimawarda.andalib@usm.my

**Abstract:** Small- and medium-sized enterprises (SMEs) in Bangladesh have been growing for the preceding two decades in various ways. Even though these enterprises exert an important effect on the national economy of the country, they face certain challenges and suffer in quite a manner. To reduce the suffering and help deal with these unavoidable challenges, SME foundations in Bangladesh have been working a lot to assist these enterprises. Usually, the challenges are financial, regulatory, or innovation performance-related. Since the enterprises are focused on particular products and their manifestation and production, performing innovatively within these organizations becomes quite a constraint factor. In this paper, a qualitative method was applied with soft systems, where an innovation model was commenced to reduce the challenges and constraints of SMEs in Bangladesh. From the literature, 50 articles were analyzed, a content and thematic analysis was done, and eight main challenges were highlighted and finalized. On the other hand, a focus group discussion among 10 entrepreneurs in Bangladesh was done, where collected data were stored in the NVIVOMac tool, transcribed, and coded, and 55 open codes were identified. These open codes were transferred to 20 axial codes and 8 themes to construct a conceptual innovation model for SMEs in Bangladesh by applying soft systems techniques to connect the themes to each other.

**Keywords:** Conceptual Innovation model; Small and medium enterprises; Bangladesh; Soft systems techniques; qualitative method; focused group discussion; challenges

## 1. Introduction

Small- and medium-sized enterprises (SMEs) are the backbone of Bangladesh's developing economy [1,2]. This sector has been a substantial protagonist in the socioeconomic development of the nation. Since the sustainable growth of SMEs can reduce poverty and unemployment in the country, it is important to understand and invent mechanisms for their sustainable growth as well [3–5]. Distinctive scholars have mentioned that SMEs are a vital tactic toward the economic liberty of any nation in the world, and they are specifically a core approach for financial independence in Bangladesh [6–10]. Therefore, the government of Bangladesh (GOB) has been considering SMEs, especially manufacturing SMEs, as a dominant industrial sector [11,12]. Nevertheless, due to various constraints and challenges, SMEs in Bangladesh cannot perform in a proper manner, nor can they generate the expected GDP, and sometimes they also get closed down [13–15].

Numerous researchs have been steered in Bangladesh to perceive the challenges faced by the SMEs of Bangladesh [15]. However, to our knowledge, none has scrutinized applying qualitative method with philosophical ontological paradigm to directly deal with the entrepreneurs in a way where the innovation model can be constructed to reduce these challenges and also where, employees' rights have been considered as a separate, distinct feature or theme. Thus, this study tries to fill the

breach and to offer possibilities of more and more future research on Bangladeshi SMEs challenges and resolutions. [16–20] This paper therefore, attempts to analyze these issues and challenges being faced by the small and medium enterprises in Bangladesh, also deals with the entrepreneurs of Bangladesh to take their opinions regarding resolutions and puts effort to create an innovation model to reduce these challenges and constraints of the manufacturing SMEs of Bangladesh. Researchers believe that the findings of this study will help SME entrepreneurs and policy-makers be prepared for challenges by understanding their strengths and weaknesses, and they will also assist entrepreneurs in converging on the constructed conceptual innovation model as per the concept of Sitwala 2014 to reduce the challenges in their SMEs [21–23].

## 1.1. Background

The progress and subsistence of small and medium enterprises are endangered by inhibitions that exist in the setups and organization's management. Diverse challenges have been identified from previous studies, which are financial instability, financial loan constraint, inexperience and lack of innovation and creativity in the field of business, regulatory licenses and taxes, increased production and manufacturing cost, poor managerial skills, issues regarding employees' rights as the key challenges hindering the success of SMEs in developing countries [24–26].

It has been observed that entrepreneurs with procedural backgrounds may find challenges in handling progressions and operative areas of the industry [27–29]. The growth of small and medium enterprises may also be obstructed by some major external challenges, such as competition, unfriendly business environment, government regulatory and law issues and the condition of the economy [30–33]. The key challenge faced by SMEs in most developing countries is meeting both local and global competition from established businesses [34]. SMEs' affordability and competitiveness in the developing countries is hindered since there is lack of trained manpower and development skills, access to adequate finance and because of increased production cost [35–37].

Regulatory and legal issues are troublesome challenges those are continuously faced by the SMEs' entrepreneurs [38]. Previous studies have also noted that the continuous urge and need to comply with the newly developed regulatory taxes, licenses are quite a headache for the entrepreneur besides the SMEs' operational factors that quite obviously hinder the success of SMEs in the developing country like Bangladesh [39]. SMEs also face adversity in dealing with employees' rights as in following the Bangladesh Labour Act 2006 (BDL) guidelines regarding employees' rights protocols because of small size, inappropriate infrastructure and workflow [40]. Previous scholars found in plenty of their works that SMEs of Bangladesh are frequently facing financial constraints, financial instability and not provided bank loans [41].

Many studies have highlighted the growth and development of the SME sector in Bangladesh. Previously, scholars have mentioned that SMEs in South Asia, particularly in Bangladesh, contribute between 40% and 60% percent of the total output or value in many South Asian developing economies, and they also account for over 70% of total employment [42,43]. Nevertheless, when the SMEs face constraints the natural tendency of resolving becomes 'cost cutting' or 'downsizing employees' that hugely affects the recruited employees at work, because most of the employees are downsized and others feel insecure of their jobs as a result the SME loses its credibility and energetic force. Therefore, entrepreneurs face more problems instead of resolving it peacefully.

## 1.2. Motivation of This Study

For last two decades, previous scholars and researchers have been doing several research works on the SMEs of distinct countries of the world. The scholars of Bangladesh have been doing research on the SMEs even more to improve this sector for the national interest of the country. For this purpose, SME foundation of Bangladesh has been built also to facilitate, e the researchers as well as the real entrepreneurs of SMEs of Bangladesh [44].

Though, a lot of research already have in Bangladesh and also some are in the ongoing process, researchers of this study feel that one of the core issues have been ignore quite frequently in these research works, which is the innovation model considering employees' rights [45]. This is high time to work and resolve the core issue else the challenges may get more complex day by day and in future there might be less opportunities to resolve and construct an innovation model. This is the prime reason to work on this present-day driven topic, and the researchers think that it will be a great initiative to provide important messages along with the innovation model to the entrepreneurs. Besides this, the study has massive rationalization from academic and real-world perspectives [46,47]. Moreover, Hoque mentioned in 2018 that there are significant research about SMEs of Bangladesh which are mostly empirical, however, more emphasize on qualitative designed research regarding SMEs of Bangladesh are required to explore and perceive direct perceptions of the entrepreneurs [27].

### 1.3. Research Objectives

In this paper the researchers have set mainly two objectives. Firstly, to observe and identify the SMEs' challenges faced by the entrepreneurs, which are driven from previous scholars' works as well as perceived from the focused group discussion. Secondly, to construct the innovation model to resolve these challenges in the SMEs of Bangladesh, hence the entrepreneurs of Bangladeshi SMEs have disclosed their opinions regarding resolutions in the focused group discussion.

### 1.4. Significance of This Study

This is a paper blending the literature and the practical scenario by using the qualitative methodology following the pathway of ontological philosophical paradigm [16,17,19,20]. This paper is specifically based on the SMEs of Bangladesh where some entrepreneurs were brought to a focused group discussion to discuss about the challenges they face in practical life. During the discussion the agendas and components brought by the entrepreneurs are listed and recorded and analyzed by NVIVOMac software. During the discussion the entrepreneurs also discussed about the probable solutions and resolution techniques of various challenges. Later, the researchers brought all of those major components together beneath one umbrella and developed a converged conceptual innovation model to resolve the challenges (the concept of convergence was taken from Andalib described in 2018) [41]. This model can be implemented in these SMEs during functioning, and the changes are expected to be positive. The researchers also brought up the issue of employees' rights, which was observed to be neglected and negative in most of the SMEs in Bangladesh: since SMEs are not absolutely organized in terms of size and operation, their main focus remains on product preparation and escalation instead of employees' welfare, rights, and innovation concepts [42,43]. Therefore, the researchers believe this paper can have a substantial impact on the SMEs and entrepreneurs of Bangladesh and can assist in their regular operation.

## 2. Literature Review

The major challenges faced by various entrepreneurs of SMEs of Bangladesh and around the world have been observed and identified from previous scholars' works. From the focused group discussion also the challenges are discussed and found by the researchers, some of which also matched with the previous scholars' works [48]. Nevertheless, the previous studies those discussed about these challenges are mentioned in this section of the paper. The common core challenges those have been found both from the previous scholars' works and from real time primary data have been disclosed here to gather and discuss previous scholars' evidences.

Scholars' in last two decades have been doing research on the SMEs challenges faced by the entrepreneurs in Bangladesh and other parts of the world face those. The strange thing is that the challenges in these two decades have remained somewhat the same with very less exceptions in real life [44–47]. The two decades are explicitly considered from firstly 2000–2010 and secondly, 2011–2019.

In Table 1, scholars' evidences regarding the challenges faced by the SMEs in two decades are presented in brief manner. Here, fifty articles (50) are mentioned in a nutshell.

**Table 1.** Scholars' evidences of challenges faced by small- and medium-sized enterprises (SMEs) over two decades.

| Challenges Faced by the SMEs | Scholars' Works from 2000 to 2010 | Scholars' Works from 2011 to 2019 |
|---|---|---|
| Financial instability (FiS) | Montoo (2006); A World Bank survey (2002) | Alauddin & Chowdhury (2015); Chowdhury et al. (2013), Hoque (2018), Hoque et al. (2016a), Hoque et al. (2016b), Hoque et al. (2018), Hoque et al. (2016c) |
| Financial loans (FL) | Mahmud (2006); Jahur & Azad (2004) | Alauddin & Chowdhury (2015); Hoque et al. (2016c); Hoque et al. (2017) Rahman (2019); Islam et al., (2013); Hoque & Awang (2016) |
| Managerial inexperience (MiE) | Stone-Romero (2003); Baron & Shane (2007); Omerzel & Antonic (2008); Islam, (2009) | Hussain & Shah (2015); Idar and Mahmood, (2011); Zaman et al., (2011); Hoque and Awang (2016), Hoque et al. (2018) |
| Lack of innovation performance (IP) | Skarzynski, P., & Gibson, (2008) | Zeebaree & Siron (2017). Zhang, & Zhang (2012) and Michelino et al. (2014); Chowdhury et al., (2013); Ibrahim (2016) |
| Regulatory licenses and taxes (RLTs) | Kazooba (2006); Martinsons (2008) | Hoque (2018); Okpara (2011); Fumo & Jabbour (2011); Chowdhury et al., (2013); Kim (2011); Khan (2012); Hoque et al. (2017); Ibrahim et al., (2016) |
| Competitive environment (CE) | Islam (2009); Montoo (2006); Mutula (2007) | Zaman et al. (2011); Urban & Naidoo (2012); Lucie (2019); Hussain & Shah (2015); |
| Increased production costs (IPCs) | Olawale (2010); Martin (2008); Mahmood (2008) | Urban & Naidoo (2012); Khan (2012); Hoque et al. (2017); Hoque et al. (2018); Ocloo (2014) |
| Employees' rights (ERs) | Martin (2008) | Andalib (2018); Hoque & Awang (2016); Muniapan (2015) |

## 2.1. Financial Instability (FiS)

One resilient SME sector contributes exceedingly to the economy, since this sector contributes to the gross domestic product, by lessening the level of unemployment, reduction in poverty levels and promotion of entrepreneurship activity [49]. A survey about the development of SMEs in Bangladesh done by the World Bank in 2002 revealed that Bangladesh SMEs' lack of finance is the critical concern. In addition, a "Report on the Contribution of Banks to the SME Sector in Bangladesh" has asserted that SMEs have always been short of business sustenance and that the entrepreneurs of SMEs receive financial assistance on a very rare basis: thus, financial instability occurs quite frequently [50].

## 2.2. Financial Loan (FL)

SMEs of Bangladesh have restricted financing attributes from banks, which is only around 10%, while self-finance remains the major source of their finance contributing 76.5% of fixed capital and 51.8% of working capital [51]. Hoque described that from his studies he observed that in Bangladesh most of the SMEs almost 89% of the SMEs obtain loan from microfinance institutions (MFIs) with higher credit interest since, Bank is not interested to help the SMEs in terms of loan. For SMEs, to construct fixed and working capital from banks is difficult since banks are not so willing to provide a loan of small size for the higher cost [52,53]. Generally, the commercial banks specify credit only to the well established and large scale trading and manufacturing customers because it is easy to handle and convenient to support them [54].

A lot of SMEs receive loans from friends, relatives and family and also break the personal savings to develop a new SME. Econometrics outcome shows that education, firm age, marital status, initial outlay, number of employees, and education do not have any influence on credit rationing. On the contrary, age and gender of the owners of the firms, heads of household, status of the living and work place and household size have impact on credit rationing. The commercial Banks desire to keep records of previous and current

income status with some deposit bond to provide loan for the SMEs, which often becomes quite impossible since the new entrepreneurs need their capital to start the SME at the very first place [55].

### 2.3. Managerial InExperiences (MiE)

Due to lack of training and business skills and a lack of managerial experience, entrepreneurs also face an inhibition in their SMEs' smooth operation. Most SME entrepreneurs follow the leadership style that comes from an informal space and from cultural norms that might be quite nonprofessional for the longevity of the firm [56]. Earlier that the behavior of the top management and the employees is always eventually the core concern and factor in sustaining any organization. Entrepreneurs must be capable of dealing with their employees and must have the knowledge and skills to motivate the employees and place them in the right positions to get the work done. Inexperience of the entrepreneurs often creates such constraints in the firms that they might just linger on and fail the organization pretty badly [57].

### 2.4. Lack of Innovation Performance (LIP)

Innovation performance inside a firm is a core component that is connected and related to other core components of the firm. In any SME the products those are offered to the customers and market by the employees but various methods and ways are there to do that. When any SME has talented and creative employees the innovation process inside the firm exist and their performances stands out as a result the specific SME stands out of all but the lack of it may hinder the process of entire operation inside the SME [58].

### 2.5. Regulatory Licenses and Taxes (RLT)

Regulatory licenses and taxes imposed by the government becomes a continuous issue for the SMEs that hinder the progress and operational process all the time [3–5,57]. SMEs are under continuous pressure to comply with new regulations, license issues, tax issues as a result the SMEs need to be very aware with it since the slight deviation might cause a huge hindrance for the SME operation [6].

### 2.6. Competitive Environment (CE)

Competitions among the SMEs sometimes trigger the organizations to produce more customized and better products and penetrate the market in a more customer oriented manner that eventually assists the progress of the SMEs [56]. Nevertheless, competitive environment has to be created and nurtured by the SMEs in such a way so that the competition can help each other rise not to destroy each other [54,58]. In Bangladesh, the SMEs are mostly formed in few specific areas like food arena, manufacturing, ceramics, garments where the competition already exists unless and until there are powerful or distinct features in the products, competition persists and it continues in such a manner that the new comers are unwelcomed.

### 2.7. Increased Production Cost (IPC)

Operations skills are reported to be deficient and are often cited as a main cause of failure in small and medium enterprises (SMEs) and when operation skills fail the production cost rise due to financial deficiency, due to lack of proper competition with the competitors and etc. Nevertheless, Production cost can also increase if there is lacking in innovation performances and technology is outdated for the production or the updated technology has to be bought from a third party outbound service from another country [59].

### 2.8. Employees' Rights (ER)

Andalib mentioned in 2018 that employees' rights must be distinctly seen as a component in any organization and it must be considered one of the core components [41]. As per Bangladesh Employees' Federation, BEF, Bangladesh Labour Act 2006 has been passed with employees' protocols those need to be addressed by all the organizations; specifically those are listed in the stock market and in the

SME foundation of Bangladesh [60]. The employee rights protocols are derived under the shadow of international labour organization's ILO protocols as well [45]. Whenever employees' rights are properly fulfilled employees automatically will be satisfied with their job, will not wish to leave the job and also will be self-motivated [60].

## 3. Research Methods

Qualitative method has been undertaken in this research with the ontological philosophical paradigm besides 'content analysis' and 'thematic analysis' in the literature part [61,62]. Almost 50 articles have been reviewed contents and themes have been identified and analyzed by using NVIVO Mac tool. Meanwhile, also focused group discussion took place. This research aims to scrutinize the core challenges of SMEs in Bangladesh by analyzing and reviewing 50 published articles and scholarly works. Data were collected from various sources using standard techniques, where this methodology can be considered as systematic qualitative analysis, since philosophical paradigms with interpretative technique has been used to summarize important information [41]. Also, by following Creswell and Auerbach & Silverstein's focused group discussion have been done in a group of 10 entrepreneurs of Bangladeshi SMEs where these core challenges have been agreed, described in detail and also solutions have been addressed [16,17]. Finally, the conceptual model has been constructed also by following the footprints of distinctive scholars [17,19]

### 3.1. Data Collection Steps

In the current study, data were collected and analyzed by following qualitative methodical steps [16,17]. Firstly, a computerized database search was performed using Proquest, Emerald, and Elsevier. Secondly, the literature search was conducted by using few specified key words like SMEs in Bangladesh, Barriers and Challenges faced by SMEs and entrepreneurs, Entrepreneurs resolution techniques, Successful entrepreneurs and etc. All articles found were stored and analyzed in NVIVOMac tool. Meanwhile, Focused Group Discussion was done in a group of ten (10) entrepreneurs and the voice records with transcriptions have been stored in NVIVOMac tool as well. These ten entrepreneurs were selected by following a selection criteria both applying snowballing and purposeful sampling method. Initially, researchers collected the list of SMEs from the SME foundation of Bangladesh. Then, three criteria were set to find successful ten SMEs and its owner for the focused group discussion. The three criteria are 1. Dhaka based SMEs, 2. Following the upgraded Company Government Act 1996 and 3. Employee Numbers < 500 as per Akhtaruddin [63] Therefore, invitation letters were sent to these SMEs' entrepreneurs, the ones who responded, were available and agreed with the consent letter participated in the focused group discussion (FGD).

### 3.2. Analyze Data

NVIVOMac serves as a reliable platform to analyze any sort of data [41]. In this study, reports or scholarly papers were stored, sorted, and analyzed through NVIVOMac. Before data analysis was conducted, several selection steps were taken. Firstly, categorizing articles by external and internal components. Secondly, highlighting the articles by key words. Thirdly, deriving the core SME components from NVIVOMac Tool by applying coding mechanism. The key words were highlighted from the transcripts of FGD and saved as raw data. These raw data has been translated sequentially to open code, axial code, themes or components by applying open coding and thematic analysis method in the NVIVOMac tool. Then, the prioritized components are highlighted, identified and finalized carefully. After the components were finalized, researchers connected and inaugurated the conceptual innovation model.

As for example, Entrepreneur, E7 said, new regulations of the govt. imposed on smes on its production location has caused lots of relocation related changes, which has increased our costs regarding the production, manpower transfer, accommodation for the employees with other benefits, setting up experienced managers at distant factory locations, environmental related changes that need time to be adjusted with and some others. This transcribed quotation is considered as the

'raw text', where the highlighted points and open codes are 'new govt. regulations', 'managers with experiences', 'factory locations', 'environment', 'production cost', 'transfer cost', 'accommodation cost', 'time', 'distance' and etc. In this manner, from every quotation of each entrepreneur, several codes have been generated. From, these open codes, the repeated codes or open codes with similar meanings are grouped together to form axial codes. Then axial codes are grouped to form themes and finally themes are grouped and coded to form 'components of the SMEs'. In this study, almost fifty-five (55) open codes are generated in the backend that is translated to twenty (20) axial codes and finally to eight (8) themes but later while constructing the theory or innovation model, seven (7) themes got derived.

## 4. Results

Categorizing and grouping the components have been done based on coding mechanism that validates and justifies data with generalizability [16,17,41]. The findings were extracted from previous scholars' articles and from entrepreneurs' perceptions of focused group discussion and then summarized systematically to assist the construction of the innovation model. Firstly, Data regarding the challenges faced by entrepreneurs are presented in Table 2. Secondly, data regarding the SMEs' challenges in various regions of the world are presented and ranged as positive, negative and mixed. Thirdly, the entrepreneurs' inputs perceived from the FGD are presented and Fourthly, the found components perceived from entrepreneurs' inputs of the FGD discussion are ranged as low, medium and high. Finally, the conceptual innovation model has been constructed by applying soft systems techniques, where the components are connected to each other [64,65].

**Table 2.** Challenges faced by entrepreneurs at Bangladeshi SMEs.

| Scholars' References (Decade: 2011–2019) | Challenges | Problems Faced | Solutions | Rate Experience (Good/Moderate/Bad) |
|---|---|---|---|---|
| Rahman (2019) | Financial instability (FiS) and financial or bank loan (BL) | Constraints in financial progress of the SMEs | Microcredit financing | Moderate |
| Hoque and Awang (2016) | Managerial inexperience (MI) | Poor work environment | Transformational leadership and managerial style, entrepreneurial attitude | Bad |
| Zeebaree & Siron, 2017 | Lack of innovation performance (LIP) | Products did not attract the customers, poor innovation space | Recruited and retained talented, creative human resources, created less turnover, motivated employees | Bad |
| Hoque (2018) | Regulatory licenses and taxes (RLTs) | Continuous pressure to comply with new regulations | Kept pace with regulatory bodies and law firms for updated solutions, SME licenses, and protocols | Bad |
| Lucie (2019) | Competitive environment (CE) | Competition with other SMEs with same products generated less income | Market R&D, created market need | Moderate |
| Urban & Naidoo (2012) | Increased production costs (IPCs) | Capability to produce became less and capex costs increased | Customized products | Moderate |
| Andalib (2018) | Employee rights (ERs) | Job dissatisfaction, no flexibility at work increased turnover | Implemented BDL, fulfilled employees' rights, created job satisfaction niche, work flexibility | Bad |

### 4.1. SMEs' Challenges and Their Impacts

Next, Table 3 presents the data extracted from published articles on the impacts of these SMEs' challenges in various regions of the world. Findings show the core challenges are financial stability, financial loan from Banks or others, innovation performances, regulatory licenses and tax issues, employees' rights, increased production and manufacturing cost and etc.

**Table 3.** Impacts of SMEs' challenges in various regions of the world

| Challenges of Bangladeshi SMEs | LR of Bangladesh | LR of Asia | Other Western LRs |
|---|---|---|---|
| Financial instability (FiS) | + | + | + |
| Bank loans (BLs) | # | + | # |
| Managerial inexperience (MI) | + | + | # |
| Lack of innovation performance (LIP) | + | # | # |
| Regulatory licenses and taxes (RLTs) | + | + | # |
| Competitive environment (CE) | ++ | # | # |
| Increased production costs (IPCs) | ++ | # | # |
| Employees' rights (ERs) | + | # | ++ |

In Table 3, Impact of SMEs' challenges in various regions of the world are analyzed in ranges. Data are presented following Miles, Huberman, and Saldana's technique where the impacts have been analyzed by ranges: Positive (++); Negative (+); Mixed (#) [66]. Next Table 4, represents the Entrepreneurs' and their current situation of their SMEs. Basically, the individual entrepreneur and in which type of SME he or she is working, when did he or she establish the SME, what kind of product the particular SME has been manufacturing and selling, the factory's location of the SME and its size. Since, the focused group discussion has been among these ten entrepreneurs from distinct different background, there whereabouts have been recorded. Mostly, the entrepreneurs' who participated in the FGD came from manufacturing kind of SME, which are mostly medium or small in size. Most of the SMEs have been established either in the year range 2000–2010 or 2011–2019, in these two decades. These entrepreneurs are thus quite established with their SMEs for quite a while but have been struggling with various constraints in their parts.

**Table 4.** FGD among 10 entrepreneurs' (their inputs).

| Entrepreneur | SME Type | Establishment Year | Products | Factory Location | SME Size |
|---|---|---|---|---|---|
| E1 | Manufacturing | 2012 | Ceramics | Gazipur | Medium |
| E2 | Manufacturing | 2008 | Paints | Narayanganj | Medium |
| E3 | Publishing–printing | 2005 | Books | Banglabazar | Medium |
| E4 | Food business | 2013 | Restaurant | Dhanmondi | Small |
| E5 | Food business | 2015 | Catering | Banani | Small |
| E6 | Manufacturing | 2000 | Cloth color | Old Dhaka | Medium |
| E7 | Garments | 2010 | Clothes | Mohakhali | Small |
| E8 | Garments | 2009 | Clothes | Uttara | Small |
| E9 | Garments | 2004 | Clothes | Dhanmondi | Small |
| E10 | Garments | 2000 | Clothes | Dhanmondi | Small |

According to the findings, financial instability (FiS) poses negative impacts on the entrepreneurs' in Bangladesh, while in Asia and in other global regions also it has negative impact. Interestingly, Bank Loans (BL) has mixed impacts and Management in experiences (MiE) has negative impacts in both Bangladesh and in Asia but they have mixed responses in western region. Innovation Performances (LIP) has negative impact in Bangladesh but has mixed impacts in both Asia and Western region. Regulatory Licenses and Taxes (RLT) have negative impacts in both Bangladesh and in Asia but has mixed responses in Western regions. Competitive Environment (CE) and Increased Production Cost (IPC) have positive impacts in Bangladesh but mixed impacts in Asia and in the Western regions. Employee Rights have negative impact in Bangladesh, mixed in Asia and positive in the western regions. Eastern influences are significantly distinct than that of western influences because of the cultural and religious dimensions [67].

In Table 5, SMEs' challenges those are observed and identified from literature review are discussed in the FGD session by following Stake (1995) and inputs are taken from the entrepreneurs in three ranges, which are either High, Low or Medium.

**Table 5.** Inputs from FGD discussion regarding challenges faced.

| Challenges | E1 | E2 | E3 | E4 | E5 | E6 | E7 | E8 | E9 | E10 |
|---|---|---|---|---|---|---|---|---|---|---|
| Financial instability (FS) | Medium | Medium | High | Low | Low | High | High | High | Low | High |
| Bank loans (BLs) | Medium | Medium | High | Low | Low | High | High | High | High | High |
| Managerial Inexperience (MI) | Medium | Medium | Medium | Medium | Medium | Medium | Medium | Medium | Medium | Medium |
| Lack of innovation performance (LIP) | Medium | High | Low | High | High | Mixed | Low | Low | Low | High |
| Regulatory licenses and taxes (RLTs) | High | High | High | High | High | High | High | High | High | High |
| Competitive environment (CE) | High | High | High | Low | High | High | High | High | Low | High |
| Increased production cost (IPCs) | High | High | High | Low | High | High | High | High | Low | High |
| Employees' rights (ERs) | High | High | High | High | High | High | Low | Low | High | High |
| | High | High | | | | | | | | |

*4.2. Innovation Model*

From the focused group discussion by following the qualitative norms of Stake, Creswell and Andalib these themes have come up also which are noted as challenges faced by the SMEs of Bangladesh. To resolve these challenges a conceptual innovation model has been constructed and proposed by the researchers in this paper that connected the themes by soft systems techniques (coding them with each other and connecting the resolution techniques to each constraint distinctly). While, developing the conceptual innovation model, the financial constraints like financial instability and Bank loan got merged into one (Figure 1).

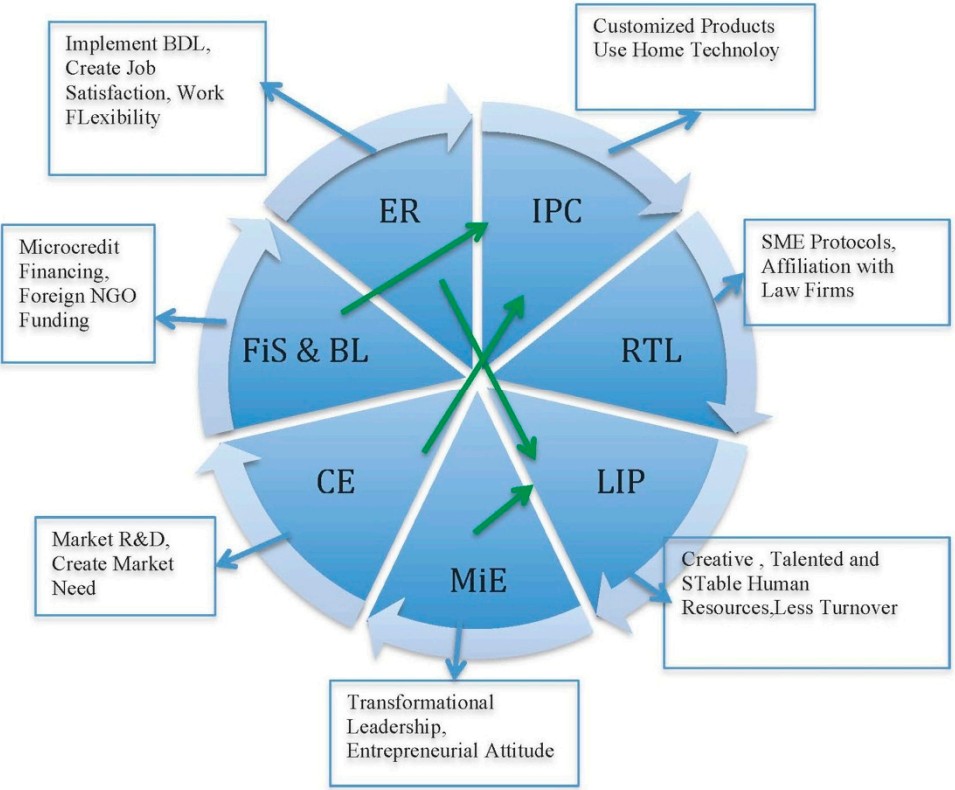

**Figure 1.** Conceptual innovation Model for the SMEs of Bangladesh.

The figure has revealed the constructed conceptual innovation model that has been proposed by the researchers in this study. Here, the components found from the FGD discussion that were actually the techniques to resolute were observed and connected to their related constraint. However, each constraint or challenge is dependent on the other and very much connected. Financial stability depends on new government regulations, and SMEs increase taxes and licenses protocols. However, each constraint or challenge is dependent on the other and very much connected. Financial stability depends on the Govt. New Regulations and SMEs increased Taxes and Licenses protocols. Employee Rights' Models can improve when the financial instability lingers on. Meanwhile, When employee Rights are fulfilled the employees' feel like home, more satisfied on job, wishes to leave the company less and entrepreneurial and innovative activities and attitude are revealed. Nevertheless, lack of innovation causes the major fallouts in the SMEs, therefore in this conceptually innovation model every major aspect has been taken take of wherever the constraints are found and specifically, the managerial and employees are considered as the core of innovation. Moreover, the talented and creative people wish to stay more in the organization and innovate new methods. Nevertheless, flexibility drives creative people more to any new organization.

Here, the connections of the challenges towards each other means, when one constraint has been resolved such as the financial instability and bank loan, other constraints involving cost issues

automatically gets resolved. The resolving technique of the financial instability and bank loan is to try to receive fund from microcredit, microfinance organizations in Bangladesh who also work with Universal Declaration of Human rights like Grameenbank Ltd., Brac bank Ltd., Association for social Advancement (ASA) foundation or other trying to gather fund from financial leasing companies like Industrial Development Leasing Company Ltd (IDLC) or from foreign distinct NGOs or individual funds from any large established organization or individual [68]. Increased production cost actually is very much inter-related with this financial instability constraint. Whenever, the financial constraint is taken care of this increased production cost automatically gets resolved somehow; however, using local technology or customizing products with fewer budgets can also solve it [54]. The resolving technique of challenge competitive environment is to research the market thoroughly, to create need of the product in the market [37,43]. To deal with the Managerial in experiences, transformational leadership style that deals with employees in more humanitarian way and entrepreneurial attitude of these leaders and managers are necessary [37,59]. To resolve the lack of innovation and performance of the employees, SMEs must stabilize the creative and talented employees by providing trainings or other job oriented benefits or safeguard their positions that will eventually cause less turnover [69,70]. This constraint is very much related to employees' rights constraint as well. Whenever, the employees' rights are taken care of and employees can view their rights and respect handled in humanitarian and legal way, they feel more safe, satisfied and content with their jobs [41]. If employees are given flexibility at work, their motivation also grows higher. One of the most core challenges is the regulatory taxes and licenses that can be resolved by complying with the protocols and affiliation with appropriate law firms in a constant manner [39].

## 5. Discussion, Limitations and Future Recommendations

Several major components are found in this paper, which reveals the impacts on entrepreneurs' performance in Bangladesh. Based on the data collected, seven challenges faced by entrepreneurs' performances have been determined. These seven components are Financial instability (FiS), Bank Loan (BL), Managerial in Experiences (MiE), Lack of innovation performances (LIP), Regulatory Taxes and Licenses (RTL), Competitive Envrionment (CE) and Increased Production Cost (IPC) and Employee Rights (ER). Researchers' have followed qualitative methods' distinct techniques of distinct scholars to analyze and explain the variables' influences and impacts [16–19,41,61,62,66,71]. By knowing the challenges of SMEs and their distinct resolving techniques in a nutsell, entrepreneurs gain insights on the aspects that can be improved at the workplace of SMEs and can deal with the challenges and resolve those one by one in a very technical manner. In addition, the constructed innovation model can be applied by the entrepreneurs' in their own SMEs to test and observe if the challenges are resolved in a partial manner or fully.

The researchers faced few challenges and limitations while conducting this study, especially before the focused group discussion. It was very difficult to bring the entrepreneurs into one table for discussing the challenges that they face in their own SMEs and the resolutions that they think it is appropriate for the SMEs. For this purpose, researchers needed to spend quite an amount of time to invoke trust in the entrepreneurs and provide them ethical guideline as in the confidentiality letter that their names and SMEs names won't be disclosed directly in any platform. Secondly, during the focused group discussion the list of challenges were too many to address by the researchers. Therefore, after the data analysis the highlighted significant challenges and their resolution techniques were disclosed and discussed with the participants again individually. After receiving their assessment and agreed version of the most of important list of challenges, those were matched with previous scholars' found evidences. The rest of the challenges got either merged or omitted for the sake of this study.

In future, scholars can implement this innovation model into various SMEs of the world and can test if any other component needs to be added or deleted from the model. Also, in future in Bangladesh the scholars can do some surveys to find out how many challenges still do exist and how many have been partially or completely resolved. Scholars can do some multiple case studies

also choosing distinct SMEs of Bangladesh to find out situation of each constraint in a more in-depth manner. There can be future studies based on these challenges to compare the present and future scenario of distinct industrial SMEs of Bangladesh.

## 6. Conclusions

Bangladesh's economy is in an emergent state, where SMEs of the country, its development and constant growth is highly necessary. This innovation model can resolve the core challenges faced by the SMEs if implemented by the entrepreneurs in their particular SMEs. Moreover, the notion of considering employees' rights have also come up in this innovation model as a resolving technique that will in general safeguard positions and jobs of many employees at the SMEs and will also provide the owners and leaders of the SMEs to think in a more humanitarian manner to deal with these challenges by keeping the SME family together under one umbrella. Because, a qualified and motivated employee who feels safe in his or her job place becomes more active, motivated and entrepreneurial in attitude that eventually helps the SMEs in the long term for sure [72]. This situation also prevents all hostile situations in any SME even in the moment of torment and gradually develops a better platform for all. However, SME components have to be connected to distinct resolution techniques and also to each other to reveal that every challenge or constraint faced by SMEs can be resolved by following a certain resolution method.

With the completion of this study, the researchers believed that the aforementioned research objectives have been achieved. The aim of this study is to enhance our knowledge regarding the challenges faced by SMEs' in Bangladesh and how these challenges can be resolved by implementing the newly constructed and proposed innovation model in Bangladesh. The novelty and originality of this research lies very much in its entire end-to-end qualitative method since Hoque also suggested in this works regarding SMEs that such qualitative research regarding SMEs' challenges and solutions needs to be done more. Moreover, the component 'employees' rights' inclusion in the conceptual innovation model has been a distinct, unique and novel addition, because employees rights issues have been always separately dealt by the management of organizations by following the Bangladesh Labour Act 2006's guideline and never really included in the organizations main domain [41]. The moment, employees' rights domain is included in the SMEs main domain, the challenges regarding it and the resolving techniques regarding it also gets emphasized as a result employees feel more safeguarded and secure.

**Author Contributions:** Conceptualization, T.W.A.; data curation, T.W.A.; formal analysis, T.W.A.; methodology, T.W.A.; software, T.W.A..; validation, T.W.A.; formal analysis, T.W.A.; investigation, T.W.A.; resources, T.W.A.; data curation, T.W.A.; writing—original draft preparation, T.W.A.; writing—review and editing, T.W.A.; visualization, T.W.A.; supervision, H.A.H.; project administration, T.W.A.; funding acquisition, H.A.H.

**Funding:** We would like to express our appreciation to Fundamental Research Grant Scheme (FRGS)—203.PMGT.6711585 under the Ministry of Higher Education Malaysia for funding this project.

**Conflicts of Interest:** The authors declare no conflict of interest.

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
