# Peer review of "Convergence of Conceptual Innovation Model to Reduce Challenges Faced by the Small and Medium Sized Enterprises’ (SMEs) in Bangladesh"

_2199-8531, doi:10.3390/joitmc5030063_

Round 1

Reviewer 1 Report

The authors attempted to identify the challenges of SMEs in Bangladesh and how entrepreneurs deal with the challenges they faced. By conducting a content analysis of previous related literature and focus group interview with ten entrepreneurs, the authors suggest seven challenges faced by entrepreneurs and how the challenges are dealt with by entrepreneurs. I see some potential in the paper, but a lot of work will be required to get it into publishable form. First, the motivation of the study is not persuasive because of weak theoretical and practical background the authors suggested in the manuscript. Second, the authors did not thoroughly review previous literature, including recently published studies in journals. Third, the methods of the study have critical problems. Lastly, because of the grammatical error, it is hard to understand the several parts of the manuscript. I genuinely appreciate the authors’ efforts in contributing to develop the research area of SMEs studies. I do hope that the authors find the comments constructive and that they consider these as an attempt to help them enhance the quality of their manuscript. I wish the authors best of luck as they pursue research in this area.

Author Response

Point 1: [1. Originality: Does the paper contain new and significant information adequate to justify publication?]

I hardly find out any theoretically and practically novel insight from the authors’ paper. In my opinion, the main reason is from the fact that the background section is too weak to explain the motivation of the study, the research objectives the authors raised. Moreover, I searched and found several similar research articles that were already published in journals. I explained this issue in the comment on Relationship to Literature in detail.

Response 1: Yes this paper contains significant originality and novelty. First of all, even though there are previous authors’ research works regarding this topic, most of the papers followed a quantitative method with a survey and did not go through qualitative method based on ontological paradigm applying focused group studies with the entrepreneurs directly discussing about their faced challenges and the resolving techniques used in their SMEs. Secondly, by taking the comment and suggestion of the reviewer 1, I have incorporated more recent studies in the literature review and have mentioned those in this study and to clear the relationship to literature in detail, more explanation has been provided. Thirdly, to justify the research objectives have been made more area specific so that no confusion arises.

Point 2: [2. Relationship to Literature: Does the paper demonstrate an adequate understanding of the relevant literature in the field and cite an appropriate range of literature sources? Is any significant work ignored?]

 The authors mentioned several articles about the challenges of Small- and Medium-sized Enterprises (hereafter SMEs) in Bangladesh. However, as I said, several research papers were already published in journals, such as Chowdhury et al. (2013), Khan et al. (2012), and Uddin & Bose (2013), which are closely related to the authors’ paper. Nevertheless, the authors did not review those articles in the paper. [*References: Chowdhury, S. A., Azam, K. G., & Islam, S. (2013). “Problems and prospects of SME financing in Bangladesh,” Asian Business Review, 2(2); Uddin, R., & Bose, T. K. (2013). “Factors affect the success of SME in Bangladesh: Evidence from Khulna City,” Journal of Management and Sustainability, 3(3); Khan, J. H., Nazmul, A. K., Hossain, F., & Rahmatullah, M. (2012). “Perception of SME growth constraints in Bangladesh: An empirical examination from institutional perspective,” European Journal of Business and Management, 4(7).]

Response 2: Taking Reviewer 1’s suggestion, I have incorporated more recent literature works in the literature review section and have mentioned the range of literature to efface arrised confusion. The two era of literature regarding SMEs’ and its challenges have been clearly mentioned in Table 1 with fifty articles’ references since, reviewer doubted the hard work of the researchers. Nevertheless, I believe, these constructive review has improved the paper only.

Point 3:

3. Methodology: Is the paper's argument built on an appropriate base of theory, concepts, or other ideas? Has the research or equivalent intellectual work on which the paper is based been well designed? Are the methods employed appropriate?]

I found several methodological issues in the authors’ research. First, the authors said that almost 50 articles were reviewed and analyzed by using NVIVO software, but the list of articles are missing. Second, in the ‘3.1. Data Collection Steps,’ the authors mentioned that the literature search was conducted by using the keywords, ‘SMEs in Bangladesh,’ ‘barriers and challenges faced by SMEs,’ ‘solutions for reducing the gaps.’ The first keyword was appropriate, but second and third keywords were not appropriate because the main objective of the analysis is to identify the SME’s challenges faced by the entrepreneurs and how they deal with those challenges in Bangladesh, not in other geographical areas. In other words, several research articles reviewed and analyzed in this study were not appropriate to the main objective of this study. I found evidence of this possibility in ‘4. Results’ section. For example, contexts of several papers mentioned in Table 1, such as Okpara (2011; Nigeria) and Fumo & Jabbour Jabbour (2011; Mozambique), are not Bangladesh. Third, how the innovation model (Figure 1) was derived is not adequately explained in the paper. For this, the authors just mentioned that they “followed the soft systems technique of Denai et al. (2007).”

Response 3: I do not quite agree with this point absolutely. Because, this paper is not just based on literature works but also on the focused group discussion, where the first challenges parts have been thoroughly reviewed from previous scholars’ distinct works. Nevertheless, I have included all the articles here below now. Secondly, content analysis have been done on various ‘keywords’ to identify the challenges of SMEs in Bangladesh, which also have termed, mentioned and verified by distinct scholars’ at different global areas at distinct times. That’s why I mentioned their references to crosscheck and match the challenges to make a list of only significantly core and important challenges. However, to support my argument I have mentioned some recent Bangladeshi scholars’ works where these challenges are also mentioned. Thirdly, the challenges faced by the entrepreneurs are found by coding method in NVIVOMAc , a systematic qualitative method followed by Auerbach & Silverstein, Yin and Eisenhardt. Then, these codes were transferred to axial codes and themes by methodical steps by following the philosophical paradigm and finally, these themes were connected to each other by following the soft systems technique of Denai, 2007 and  checkland, 2010 . Nevertheless, Table 1 has been introduced with fifty articles explicitly with scholars’ references.

Point 4: [4. Results: Are results presented clearly and analysed appropriately? Do the conclusions adequately tie together the other elements of the paper?]

The Results section is superficial and largely descriptive. The data analysis process presented in the manuscript is too simple to understand how the results are derived from the raw data. Furthermore, several articles cited in the paper are missing in the References section, such as Mahmood (2001) in Table 1.

Response 4: I believe, analysis, result and conclusion sections are very much tied to each other and since it is a partial qualitative research. The result should be descriptive surely. But, it is not superficial or vague. To make it more understandable to the reviwers and readers- I have modified the paragraphs. The entire end-to-end coding method from raw data to final outcome has been already mentioned. However, I have tried to make it more transparent as possible. A qualitative outcome can have subjective interpretations being based on Ontological paradigm. Hopefully, this time it would be understood more clearly. The references, where minor mistakes were observed have been solved.

[5. Practicality and/or Research implications: Does the paper identify clearly any implications for practice and/or further research? Are these implications consistent with the findings and conclusions of the paper?]

The authors mentioned several implications of the study in the Conclusion and Discussion section. However, the implications are almost the same as the results of the study. The suggestions for entrepreneurs or policymakers based on the innovation model (Figure 1) are also superficial and are not tightly tied to the results of the study.

-3-

Response 5:The implications are not exactly the same as discussed in conclusion and discussion. However, I have made more efforts to explicitly understand the implication in a better and transparent manner. Have separated the discussion, limitation and future work section from the conclusion section.

[6. Quality of Communication: Does the paper clearly express its case, measured against the technical language of the field and the expected knowledge of the journal's readership? Has attention been paid to the clarity of expression and readability, such as sentence structure, jargon use, acronyms, etc.]

Several parts of the manuscript are hard to understand because of grammatical error. For example, I found out an incomplete sentence in the first paragraph of the Results section (Thirdly, the …). I recommend the authors to thoroughly review and revise the whole manuscript.

Response 5: Even though, as authors’ we pay our full concentration while writing down a paper, there might be nitty gritty mistakes and slight issues that can be considered. However, I have gone through the entire paper few times and also have modified few things and corrected few grammatical errors for more understanding and clarity before submitting the next version. I guess, reviewer would consider these.

[Comments and Suggestions forAuthors]

The authors attempted to identify the challenges of SMEs in Bangladesh and how entrepreneurs deal with the challenges they faced. By conducting a content analysis of previous related literature and focus group interview with ten entrepreneurs, the authors suggest seven challenges faced by entrepreneurs and how the challenges are dealt with by entrepreneurs. I see some potential in the paper, but a lot of work will be required to get it into publishable form. First, the motivation of the study is not persuasive because of weak theoretical and practical background the authors suggested in the manuscript. Second, the authors did not thoroughly review previous literature, including recently published studies in journals. Third, the methods of the study have critical problems. Lastly, because of the grammatical error, it is hard to understand the several parts of the manuscript. I genuinely appreciate the authors’ efforts in contributing to develop the research area of SMEs studies. I do hope that the authors find the comments constructive and that they consider these as an attempt to help them enhance the quality of their manuscript. I wish the authors best of luck as they pursue research in this area.

Response regarding Comments and Suggestions:

Thanks for the appreciation regarding the hard work. Indeed, bringing ten entrepreneurs under the same umbrella has been a very difficult job itself and making them talk about the challenges and the resolutions for these challenges according to their believe has been also the hard part. Connecting it with the literature and bringing the true and novel insight has been also quite a hard work. Hoping to receive a positive feedback from the reviewer this time, since few things have been arranged as per the comments and some portions I have provided logical explanation of why some things need to remain the same. It’s an honour to receive comments that are helpful for this paper. Thank you loads. It would be great to get this paper accepted and published in this journal. Thank you.

To Provide the following answers of these questions 

Does the introduction provide sufficient background and include all relevant references? 

(Yes, in the modified manuscript, I have tried to put the sufficient background information with all relevant references)

Is the research design appropriate? 

(Yes, have explained the research design more thoroughly)

Are the methods adequately described?

(Yes, the method has been described in detail manner even by putting example of the background tedious work in steps )

Are the results clearly presented?

(Yes, the results have been clearly identified and presented. Have put efforts to make the result more understandable and visible. Hopefully, this time this version of manuscript will help.)

Are the conclusions supported by the results?

(Yes, the conclusions, discussions and recommendations are reflecting the results in distinct manners and therefore, supporting each other.)

NOTE: Please Read this version of manuscript to see the major changes that have been done as per your feedback and comments. Hopefully, this time, the manuscript will create more understanding and readability . Looking forward to receive your positive feedback. 

Reviewer 2 Report

The article is interesting and describes an important research problem. However, I recommend making a few changes and additions. The main results of the research should be presented in the Abstract. It is recommended to formulate a research hypothesis. Author/s should show the originality and contribution into science in the Introductiona and the Conclusions. The focus research procedure should besedcribed. It would be an advantage to develop the discussion and separate it in a separate chapter. The literature must be improved.

The proposed changes will increase the value of the article, and after these changes the article should be published.

Author Response

Point 1:

Comments and Suggestions for Authors : The article is interesting and describes an important research problem. However, I recommend making a few changes and additions. The main results of the research should be presented in the Abstract. It is recommended to formulate a research hypothesis. Author/s should show the originality and contribution into science in the Introduction and the Conclusions. The focus research procedure should be described. It would be an advantage to develop the discussion and separate it in a separate chapter. The literature must be improved.

The proposed changes will increase the value of the article, and after these changes the article should be published.

Response 1: Thank you for your positive feedback so much. Have modified the literature section introducing a table also with 50 scholars’ references and works, those have helped this study to build up the literature data. The originality and contribution sections are again re-written. Also, the focus has been elaborated, discussion, limitations and future recommendations are discussed separately. The main result has been discussed in the abstract as well. Hopefully, the next version will provide more insight and transparency about the research to accept it for publication. Thanking the reviewer so much.

To Provide the following answers of these questions 

Does the introduction provide sufficient background and include all relevant references? 

(Yes, in the modified manuscript, I have tried to put the sufficient background information with all relevant references)

Is the research design appropriate? 

(Yes, have explained the research design more thoroughly)

Are the methods adequately described?

(Yes, the method has been described in detail manner even by putting example of the background tedious work in steps )

Are the results clearly presented?

(Yes, the results have been clearly identified and presented. Have put efforts to make the result more understandable and visible. Hopefully, this time this version of manuscript will help.)

Are the conclusions supported by the results?

(Yes, the conclusions, discussions and recommendations are reflecting the results in distinct manners and therefore, supporting each other.)

NOTE: Please Read this version of manuscript to see the major changes that have been done as per your feedback and comments. Hopefully, this time, the manuscript will create more understanding and readability . Looking forward to receive your positive feedback. 

Round 2

Reviewer 1 Report

I appreciate your significant efforts to revise the manuscript. The manuscript was improved a lot. I have an additional comment for you: *Please check the references you cited again in the paper. For example, you mentioned in the manuscript "... (Hoque et al., 2019..." (p. 3), but I cannot find the reference in the References section. Is it the reference published in 2018 or earlier?

Author Response

Response to Reviewer 1 Comments

ROUND 2

Comment from Round 2: I appreciate your significant efforts to revise the manuscript. The manuscript was improved a lot. I have an additional comment for you: *Please check the references you cited again in the paper. For example, you mentioned in the manuscript "... (Hoque et al., 2019..." (p. 3), but I cannot find the reference in the References section. Is it the reference published in 2018 or earlier?

Open Review

English language and style

( ) Extensive editing of English language and style required 
( ) Moderate English changes required 
(x) English language and style are fine/minor spell check required 
( ) I don't feel qualified to judge about the English language and style

Answer :

Thank you Dear reviewer for your feedback, comments and appreciation for this hard work. I have checked all the references and citations again. Hoque et al., 2019 in p3 was a mistake –Thank you, it should be 2018. Have corrected it. The manuscript has been cross-checked by a native English-speaking proofreader also. Hopefully, this time you would like to accept my manuscript for publication. And, have checked in Turnitin, where plagiarism shows 6% similarities only. Nevertheless, please let me know if there is anything else that needs to be corrected or modified.

Note: ‘What does the first point ‘Signing my review report’ exactly mean?' Is it a positive thing that you will not sign my report or a negative thing? Hopefully, the manuscript will be considered for further publication process now. Looking forward to hear from you.
